# NSW-EPNews: A News-Augmented Benchmark for Electricity Price Forecasting with LLMs

## Abstract

Electricity price forecasting is a critical component of modern energy-management systems, yet existing approaches heavily rely on numerical histories and ignore contemporaneous textual signals. We introduce NSW-EPNews, the first benchmark that jointly evaluates time-series models and large language models (LLMs) on real-world electricity-price prediction. The dataset includes over 175,000 half-hourly spot prices from New South Wales, Australia (2015–2024) and curated market-news summaries from WattClarity. We frame the task as 48-step-ahead forecasting, using multimodal input, including lagged prices, vectorized news for classical and state-of-the-arts time-series forecasting models, and prompt-engineered structured contexts for LLMs. Our datasets yields 3.6k multimodal prompt-output pairs for LLM evaluation using specific templates. In our comprehensive benchmarks, we identify that news features yield marginal benefits at best and can even degrade performance across traditional statistical, machine learning, deep learning and state of the art time series forecasting models. This pattern holds for open and closed-source LLMs, including ChatGPT-4o, Gemini 1.5 Pro, Meta-Llama-3-8B-Instruct, Mistral-7B-v0.1 and Qwen-2.5-7B-Instruct. It also leads to frequent hallucinations in some closed-source models, such as fabricated or malformed price sequences. NSW-EPNews provides a rigorous testbed for evaluating grounded numerical reasoning in multimodal settings, and highlights a critical gap between current LLM capabilities and the demands of high-stakes energy forecasting.

## 1 Introduction

Time-series forecasting is central to modern infrastructure, and electricity-price prediction is its canonical, high-stakes application. Accurate and transparent price outlooks steer load balancing, dispatch planning, and wholesale trading across deregulated markets (Cornell et al., 2023). Classical econometric tools—ARIMA and its seasonal extension SARIMA—remain popular thanks to their interpretability, yet their linear assumptions limit fidelity to long-range, nonlinear structure in prices (Box and Jenkins, 1994; Gonzales et al., 2024). Conventional machine-learning regressors (LR, RF, SVR) capture richer covariate effects, but still falter when the signal is buried in complex temporal dependencies or interleaved with unstructured context such as news headlines (Ballı, 2021; Ahmed et al., 2021; de Fortuny et al., 2014). In contrast, transformer-based large language models (LLMs) have recently demonstrated strong zero-/few-shot competence on time-series tasks by treating numeric sequences as another modality of "language" and conditioning on heterogeneous side information (Liu et al., 2025). Their capacity to ingest both structured features and free-form text—e.g., market commentary and temperature forecasts—raises the prospect of materially improved electricity-price prediction.

**Motivation.** Electricity prices are driven by a confluence of factors that extend far beyond historical values. Market commentary and policy news shape expectations and trading behaviour (Lazarczyk, 2016; Rogmann et al., 2024; Wei et al., 2024). Harnessing such exogenous signals therefore promises improved forecast accuracy; When information from multiple sources—such as weather data and news content—is integrated, the natural language processing capabilities of LLMs become particularly advantageous.

**Gap.** In recent years, time-series forecasting evaluation has coalesced around public benchmarks—spanning traffic, energy load, macro-economic, and weather datasets—with error metrics like MSE and MAE (and variants) plus significance testing to compare models (Zeng et al., 2022; Lago et al., 2021). In electricity-price forecasting, open datasets covering multiple regional markets and year-long test sets against state-of-the-art baselines have become standard (Lago et al., 2021). Yet these benchmarks overlook unstructured news data, despite evidence that external text can enhance numerical prediction (Andrei et al., 2024). With large language models (LLMs), zero-shot forecasting and multimodal fusion show great promise—for example, GPT4MTS's integration of news events and numerical series

yields significant accuracy gains (Jia et al., 2024). To advance the field, a benchmark that fuses electricity-price time series with news text is both novel and necessary—but must be paired with rigorous prompt engineering and fine-tuning to mitigate LLM hallucination risks (Jin et al., 2024). Despite the intuitive appeal of text-enriched forecasting, the community lacks a systematic benchmark for assessing how well LLMs integrate unstructured information into price predictions. Whether LLMs can reliably translate news sentiment and weather cues into numerical forecasts—*and whether their outputs can be trusted in high-stakes markets*—remains an open question (Liu et al., 2025; Chen et al., 2024).

Our contribution can be summarized as follows:

- **Dataset Construction:** We introduce NSW-EPNews, a large-scale multimodal dataset and benchmark for electricity price forecasting that integrates historical electricity price time series and relevant news articles from New South Wales (NSW, Australia) covering 2015–2024. This dataset provides a comprehensive testbed for analyzing how rich multimodal inputs (weather and news) influence electricity price dynamics over nearly a decade of real-world data.

- **Prompt Template Design:** We design a prompting methodology that enables both classical time-series forecasting models and large language models (LLMs) to ingest and make predictions from the combined multimodal data. In particular, we craft input templates that present the numerical electricity price data and textual news information in a unified format, allowing models of different types to effectively utilize the heterogeneous inputs and enabling direct comparison of traditional forecasting approaches against LLM-based methods on the same tasks.

- **Evaluation Framework:** We develop an evaluation framework to rigorously assess model performance, quantifying both standard forecasting accuracy and hallucination-related failure modes. This framework computes conventional prediction error metrics for price forecasts while also detecting and measuring any hallucinated content in model outputs, thereby evaluating each model's predictive accuracy alongside the factual consistency and reliability of its generated explanations or reports.

- **Key Findings:** Extensive experiments show that (i) statistical, machine-learning, deep learning, and time-series baselines gain little from the news vectors; in most cases, they perform worse when the news features are included and (ii) state-of-the-art closed and open-source LLMs can incur large errors (MAE > 70 AUD/MWh on some prompt settings) and frequent hallucinations. Open-source LLMs still struggle to capture irregularities in real-world data. Current prompting and model architectures are therefore insufficient for robust, news-aware electricity-price forecasting, underscoring the need for improved prompt engineering, retrieval grounding and fine-tuning strategies.

## 2 BACKGROUND

**Traditional Time Series Forecasting vs LLMs.** Traditional time-series forecasting has long been dominated by statistical models and machine learning methods. Classical models like ARIMA rely on assumptions of linearity, stationarity, and carefully tuned seasonal heuristics(Jin et al., 2024). These models perform well for small-scale data with stable patterns but struggle as data complexity grows. In recent years, deep learning approaches (e.g. LSTMs and Transformers) have become state-of-the-art on many forecasting benchmarks, demonstrating the ability to learn non-linear and long-term dependencies directly from data (Yang et al., 2024). Yet even advanced deep models often require structured inputs and feature engineering, making it difficult to incorporate unstructured information (such as free-form text or images). They can still miss hidden complex patterns in large-scale, diverse sequences (Yang et al., 2024), highlighting an opportunity for more flexible frameworks. Large Language Models (LLMs) have emerged as a promising new avenue for forecasting, inspired by their success in capturing complex patterns in natural language. LLM-based models leverage extensive pre-training on vast corpora to integrate broader contextual knowledge and act as general pattern learners beyond just language(Andrei et al., 2024). Initial studies have only begun exploring LLMs for time-series tasks; for example, TimeGPT and other LLM forecasters have demonstrated competitive accuracy in certain conditions, though their advantages vary with market volatility. Notably, the energy domain presents an open opportunity – until recently no LLM had been applied to electricity price forecasting, motivating investigation into whether such models can be adapted to improve predictive performance(Andrei et al., 2024). By harnessing pre-trained knowledge and zero-shot capabilities, LLM-based forecasting methods aim to overcome the rigidity of traditional models, especially in handling heterogeneous and unstructured data sources.

**External Influencing Factors in Electricity Price Forecasting.** Electricity prices are driven by a multitude of external factors beyond the historical price series itself. Informational and socio-economic factors play a critical role. News events, policy announcements, and market sentiment can all sway trader expectations and induce price movements.

Recent research has demonstrated the value of incorporating textual data: for example, leveraging GPT-based analysis of energy news can produce features that align closely with subsequent price fluctuations(Menéndez Medina and Heredia Álvaro, 2024). Such findings indicate that news articles and expert reports often contain leading indicators of market dynamics, from geopolitical developments to technical disruptions. By converting unstructured inputs (like news or social media sentiment) into quantitative signals(Menéndez Medina and Heredia Álvaro, 2024), forecasters can capture effects that traditional time-series models would otherwise miss. Overall, the evidence is clear that external influencing factors – whether exogenous physical drivers like weather or informational drivers like news – materially affect electricity prices, underscoring the importance of models capable of ingesting these heterogeneous data sources.

**Hallucination and Prompt Engineering.** LLM-based forecasters frequently generate implausible trajectories—spikes, offsets, or flat lines—that are unsupported by the input data, compromising trust in high-stakes settings (Bang et al., 2023). Recent work mitigates this by grounding the prompt with retrieved, similar subsequences: retrieval-augmented generation (RAG) constrains outputs to historical patterns and demonstrably curbs hallucinations (Lewis et al., 2021). Complementary progress comes from prompt engineering. Zero-shot instructions test a model's native generalisation, whereas few-shot templates supply a handful of input–output exemplars, yielding sizeable accuracy gains via in-context learning (**?**). Further, chain-of-thought prompting elicits intermediate reasoning steps that boost multi-step prediction quality (Wei et al., 2023a). Together, retrieval grounding and carefully designed prompts represent the current best practice for deploying LLMs in time-series forecasting while keeping hallucinations in check.

## 3 NSW-EPNews Benchmark Design

### 3.1 Data Preprocess

**Scrape news.** We scraped ten years of news articles gathered from WattClarity(Global-Roam Pty Ltd, 2025). WattClarity is established in 2007, it provides detailed analysis and regularly publishes news related to the operation of the Australian NEM, making it a trusted and authoritative source of electricity market information. To illustrate the practical relevance of integrating news into forecasting, we highlight several real scenarios where electricity market news significantly influenced price dynamics (See appendix F). Below is an algorithm that demonstrates how our scrape script works.

---

**Algorithm 1:** News scraping process

---

**Input:** `start_url`
current←`start_url`;;
**while** *(soup←get_soup(current))≠null* **do**                              // crawl archive
    **foreach** *link ∈* `fetch_news_links_from_page`*(soup)* **do if** *(art←get_soup(link))≠null* **then**
        (title,author,raw_date,date,topic)←`extract_information`(news);
        content←`classify_content`(`extract_content`(news));
        append_row_to_csv(title,author,date,topic,content);
    ;
    current←`get_next_page_url`(soup); `sleep`(shortInterval);

---

**Classify news.** Electricity-market reports are often long and heterogeneous; directly feeding the raw text into a forecasting model obscures which portions of the narrative actually matter for prices. To convert these articles into machine-parsable signals we decompose the instruction to GPT-4o into four modular blocks, each targeting a specific weakness observed in baseline LLM processing. Applying the four-block prompt,including Role assignment, Classification criteria, Key attributes, Summary rules to the entire corpus yields a concise paragraph plus structured metadata for each of the ∼3.6 k articles. This dual summarisation–annotation step serves two purposes: (i) it *compresses* lengthy documents so that LLM forecasters can attend to salient facts without exceeding context limits, and (ii) it *labels* every sample with an impact level, allowing us to probe whether models differentially exploit high-impact versus low-impact news.(More details in appendix F)

**Price data preprocess.** Our benchmark includes ten years of Australia New South Whales' electricity price data collected from the Australian National Electricity Market (NEM)(Australian Energy Market Operator, 2025). The data frequency of electricity price data recording were thirty-minutes until 1st October 2021, NEM changed the frequency to five-minutes. To ensure consistency across the entire dataset, we applied a down-sampling method to the post-2021 data. This method aggregats every six 5-minute records into a single thirty-minute interval using the median value to preserve

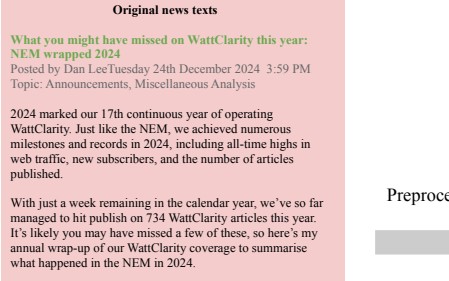

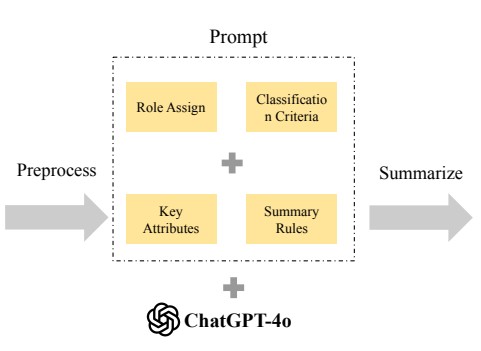



**Original news texts**

**What you might have missed on WattClarity this year: NEM wrapped 2024**
Posted by Dan Lee Tuesday 24th December 2024 3:59 PM
Topic: Announcements, Miscellaneous Analysis

2024 marked our 17th continuous year of operating WattClarity. Just like the NEM, we achieved numerous milestones and records in 2024, including all-time highs in web traffic, new subscribers, and the number of articles published.

With just a week remaining in the calendar year, we've so far managed to hit publish on 734 WattClarity articles this year. It's likely you may have missed a few of these, so here's my annual wrap-up of our WattClarity coverage to summarise what happened in the NEM in 2024.

**Year in Review**
January began with a bang. On January 22nd, the NEM's most northern region smashed its peak demand record, when market demand in Queensland exceeded 11,000 MW for the first time and surpassed the previous record by nearly 900 MW. The peak demand record coincided with temperatures in the state reaching the mid-to-high 40s.

In February, destructive storms in Victoria caused extensive damage to the transmission and distribution network, leading to power outages for hundreds of thousands of customers. Notably, six transmission towers collapsed north of Geelong due to severe winds, and all four units at ......(more texts)

**Summarized news text**

The summary of the news "What you might have missed on WattClarity this year: NEM wrapped 2024" posted at 24-12-2024 03:59:00 PM is: This article provides a comprehensive review of key events that shaped the National Electricity Market (NEM) in 2024 as reported by WattClarity. The year saw significant milestones in demand records, infrastructure challenges due to weather events, system incidents, policy announcements, and market dynamics across the regions of Queensland, Victoria, Tasmania, New South Wales, and South Australia. Noteworthy incidents include record peak demands, power outages due to storms, a critical wind drought, and significant policy announcements impacting the market. The analysis highlights essential insights into the NEM's performance and external references and appearances by WattClarity in 2024. Its relevance level is: Level 1. Its category is: Extreme weather, turbine/generator failures, grid/system instability. Its Timeframe of the event impact on the electricity_market is: Throughout 2024. Its root_cause is: Extreme weather events such as storms and wind droughts disrupting power transmission and generation. Its accident scale is: Regional with major implications across the NEM. Its dates are: Various dates throughout 2024. Its affected region is: Queensland, Victoria, Tasmania, New South Wales, South Australia. Its affected key users are: General electricity consumers across impacted regions. Its cause type is: Weather-induced incidents and grid instability. Its causes are: Severe weather conditions, transmission tower collapses, tripping power stations, grid suspensions.

Figure 1: Prompt used for news classification

---

**Algorithm 2:** Downsample 5→30-min electricity prices

**Input:** records sorted by timestamp
**Output:** 30-minute data
cutoff ← `2021-10-01 00:00`
output ← [r for r in records if r.time < cutoff]
post ← [r for r in records if r.time ≥ cutoff]
**for** *window of each 6 consecutive r in post* **do**
    $t_0$ ← window[0].time
    $v$ ← `median`({$r.value$})
    output.append(($t_0, v$))

**return** output

---

**Algorithm 3:** Simplified forecasting pipeline

**Input:** Prices, classified news
**Output:** Baseline metrics, LLM metrics
prices ← `load("prices")`
news ← `load("news")`
raw ← [ ]
**foreach** *day* **do**
    raw.append({prices: prices[day], news: `summarize`(news,day),
**foreach** *model ∈ {ARIMA, XGB, LR}* **do**
    baseline[model] ← `train_eval`(model, raw)
prompts ← `format_prompts`(raw)
**foreach** *p ∈ prompts* **do**
    llm_results.append(`call_llm`(p))

**return** {baseline, llm_results}

---

the central trend while reducing the influence of short-term fluctuations. The preprocessed data are fed to statistical, machine learning, and state of the art time series models. Because these models cannot ingest text, we convert news articles into 50 dimensional TF-IDF vectors. We then concatenate these vectors with price features to test each model's ability to use news information.

## 3.2 DATA PREPARATION FOR LLMs

**Raw JSON Dataset construction.** Records in raw dataset are stored in JSON format. Each record contains four key components: (1) the electricity price data for a single day (48 half-hourly data points), (2) the summarized news articles posted on that same day, and (3) the electricity price data for the following day (also 48 data points), which serves as the ground truth for forecasting. By doing this, each record contains one day of historical electricity price data and the true prices for the following day as the prediction target. The next record uses the previous day's true prices as its historical input. This creates a sliding window structure where one day of data is used to forecast the next day, allowing continuous day-by-day

**Raw Dataset**

```
[{
    "Historical electricity price data": ["49.85","53.57","49.79","47.99","47.22", ... (44 more prices) ],
    "Start date": "04/01/2020 00:00:00",
    "End date": "04/01/2020 23:30:00",
    "Prediction start date": "05/01/2020 00:00:00",
    "Prediction end date": "05/01/2020 23:30:00",
    "Data frequency": "30 minutes",
    "Region": "NSW1",
    "News during this period of time": ["The summary of the news \"More details on the bushfire-driven extremes in the NSW Region of the NEM on Saturday 4th January\" posted at 04-01-2020 07:51:00 PM is: ......(more texts)", "The summary of the news \"Bushfires under interconnectors through Snowy Mountains cause separation of NSW from VIC region\" posted at 04-01-2020 03:50:00 PM is: ......(more texts)"],
    "Prediction": ["57.95","54.89","57.95","54.7", ..... (44 more prices)]
},
......(more records)
]
```

Figure 2: Example record in the raw dataset

evaluation. The raw dataset will eventually have more than 3600 pairs of records. Its role is to assist building the prompt–target pairs for LLM input and evaluation. There is an example record shown in Figure. 2.

**LLM Prompt Construction.**    To test the performance of LLMs, we further preprocess the raw dataset into a new format. In this dataset, each record includes a constructed prompt and a corresponding sequence of true prices as the prediction targets, forming a complete input–output pair for evaluating the LLM's forecasting performance. Each pair is still saved in JSON format, same as raw dataset. This setup allows for a direct comparison between the model's predicted results and the ground truth values using standard evaluation metrics. It follows the same sliding window setting as the raw dataset, using one day of historical price data and background information to predict the next day's prices. Although LLMs are highly capable of processing natural human language, they can also suffer from issues such as hallucination(Ji et al., 2023), generating outputs that are fluent but factually incorrect. A single prompt format is insufficient to reliably guide LLMs toward producing accurate and reasonable forecasts(Meskó, 2023). Therefore, in our benchmark, we design and evaluate four different prompt formats. These formats vary in two key dimensions: whether they are zero-shot or few-shot, and whether they include a chain-of-thought(Wei et al., 2023b) reasoning process or not. The information in the raw dataset was extracted and reorganized to create four groups of distinct datasets, each corresponding to one of the four prompt styles (See appendix C): zero-shot, zero-shot with chain-of-thought, few-shot, and few-shot with chain-of-thought. The aim is to apply these prompting techniques to enhance the prediction performance of LLMs and to explore how their behavior varies across different prompt settings. This allows us to analyze the models' ability to incorporate news information into forecasting from multiple perspectives. To better illustrate the effects of prompt engineering and background information, we will compare LLM performance under four prompt settings against a no-news baseline to conduct an ablation study. LLMs are expected to receive prompts that integrate 48 historical price points from a single day, corresponding news summaries, time features, and to output a sequence of 48 forecasted price values for the following day in the format we have defined. For more details about the different prompt settings, please see the appendix A.

Table 1: Prompt Format Comparison Across Input Features and Design Strategies

| Prompt Type | Price History | News | Q&A Examples | CoT Reasoning |
|---|---|---|---|---|
| Zero-shot | ✓ | ✓ | ✗ | ✗ |
| Few-shot | ✓ | ✓ | ✓ | ✗ |
| Zero-shot + CoT | ✓ | ✓ | ✗ | ✓ |
| Few-shot + CoT | ✓ | ✓ | ✓ | ✓ |
| Ablation Study | ✓ | ✗ | ✗ | ✗ |

### 3.3 CLOSED-SOURCE LLMS HALLUCINATIONS AND ERROR DETECTION

We introduced supplementary measures to assess the reasoning quality of closed-source LLM-generated forecasts. Since LLMs are known to suffer from hallucinations and false responses, during our experiments with closed-source LLMs, we observed several instances where their outputs were illogical or inconsistent with the given inputs. These behaviors indicate that the model may not be reasoning about future prices based on a true understanding of the provided structured and unstructured information. To detect such cases, we implemented additional checks during evaluation (See Algorithm 3 below). Specifically, we compare the closed-source LLM's predictions with the historical input data and record the number of occurrences where direct copying, offset-based generation, or repetition patterns are observed. The rate of these hallucinations and errors is calculated by dividing the number of occurrences by the total number of prompts processed by the LLMs. In our benchmark, we are trying to detect four different types of hallucination and error outputs. They are **echoing failure**, **trivial transformation**, **degenerate copy** and **format violation** (please see appendix D).

## 4 RESULTS

**Standard Evaluation Metrics.**    We evaluate forecasting performance using four standard metrics: MSE and MAE. To account for irregular output lengths from LLMs, any extra predicted values are discarded, and missing values are temporarily filled with nulls and excluded from metric calculations. MSE highlights large errors by squaring deviations, making it sensitive to outliers. MAE captures the average absolute error in dollars and is less affected by extreme values. (See appendix G for more details). To ensure a fair comparison with the other models in this study, we applied the inverse transform and computed all metrics on the original price scale.

---

**Algorithm 4:** Hallucination and error output detection pseudocode

---

**Input:** history, K
**Output:** format_violation, echoing_failure, trivial_transformation, degenerate_copying
**foreach** *each LLM API call* **do**
    Try to parse the model's reply into a list `pred`
    **if** *parsing fails* **then**
        `format_violation` ← true
        **continue**

    *echo_count* ← #values in `pred` matching any in `history`
    **if** *echo_count* $\geq 10$ **then**
        `echoing_failure` ← true

    *offsets* ← { pred[i] − history[i] | i = 1..K }
    **foreach** *offset in offsets* **do**
        *match_count* ← #{j | pred[j] = history[j] + offset}
        **if** *match_count* $\geq 20$ **then**
            `trivial_transformation` ← true
            **break**

    *freq_map* ← frequency map of values in `pred`
    **if** *any value's freq* $> 5$ **then**
        `degenerate_copying` ← true

    Record the four boolean flags for this API call

---

**Model Used.** Baselines include SARIMAX, Random Forest, LSTM(Hochreiter and Schmidhuber, 1997), and state-of-the-art time series models: TimeMixer(Wang et al., 2024a), TimesNet(Wu et al., 2023), TimeXer(Wang et al., 2024b), and PatchTST(Nie et al., 2023). Besides price-only tests, these models also use a 50-dimensional TF-IDF news vector as multimodal features. For fair comparison with LLMs, which use one day (sampling frequency is 30 minutes, hence 48 datapoints) of history to predict the next day, baselines are evaluated with their sliding window's stride = 1 and stride = 48.

For both of the selected closed-source and open-source LLMs, including ChatGPT-4o(OpenAI, 2024), Gemini 1.5 Pro(Reid et al., 2024), Meta-Llama-3-8B-Instruct(AI@Meta, 2024), Mistral-7B-v0.1(Jiang et al., 2023) and Qwen-2.5-7B-Instruct(Team, 2024), they are tested using five prompt engineering strategies: zero shot, few shot, zero shot with chain-of-thought (CoT), few shot with CoT and ablation study. Prompts are submitted to closed-source LLMs through iterative API calls, and generated outputs are evaluated for forecasting accuracy using MAE and MSE. Additionally, outputs are examined for hallucinations, including format violations, direct copying of historical prices, trivial transformations such as constant offsets, and degenerate copying where a single value is repeated throughout. By contrast, open-source LLMs are fine-tuned and evaluated by MAE and MSE as well. This comprehensive evaluation provides a clear comparison between traditional forecasting models and modern LLM-based approaches. (Experiment settings: appendix E)

## 4.1 BASELINES AND LLMs RESULTS

Table 2: Baseline models' performance, ▾indicates improvement in results after increasing stride size, ▴indicates worse results

| Models | Electricity Price | | | | With 50-TF-IDF news vectors | | | |
| | Stride = 1 | | Stride = 48 | | Stride = 1 | | Stride = 48 | |
| | MSE | MAE | MSE | MAE | MSE | MAE | MSE | MAE |
|---|---|---|---|---|---|---|---|---|
| SARIMAX | 522449.897 | 87.532 | 477957.310 ▾ | 84.236 ▾ | 516962.132 | 87.367 | 780992.609 ▴ | 108.031 ▴ |
| Random Forest | 207472.724 | 65.759 | 232346.160 ▴ | 74.378 ▴ | 220005.819 | 70.073 | 223198.027 ▴ | 74.338 ▴ |
| LSTM | **184355.164** | **53.977** | **186491.115** ▴ | **72.272** ▴ | **187373.364** | **54.863** | **189332.058** ▴ | **64.728** ▴ |
| TimeMixer | 195509.203 | 64.728 | 218554.609 ▴ | 73.969 ▴ | 208713.172 | 61.341 | 217606.344 ▴ | 73.998 ▴ |
| TimesNet | 208088.125 | 65.715 | 232091.781 ▴ | 78.685 ▴ | 217206.641 | 69.994 | 250649.750 ▴ | 84.495 ▴ |
| TimeXer | 203066.656 | 60.089 | 234809.594 ▴ | 79.001 ▴ | 203789.875 | 59.423 | 236026.859 ▴ | 79.311 ▴ |
| PatchTST | 209318.531 | 59.154 | 235771.938 ▴ | 80.740 ▴ | 207297.938 | 65.475 | 233728.422 ▴ | 83.848 ▴ |

**Baselines.** Baseline models were evaluated with and without 50-dimensional news vectors under sliding-window strides of 1 and 48. According to Table 2, the inclusion of news vectors did not yield consistent gains and at times

degraded accuracy. With stride = 1, after running on datasets without and with news vectors, SARIMAX showed small reductions in both MSE and MAE. TimeMixer and TimeXer exhibited lower MAE but slightly higher MSE. Random Forest, LSTM and TimesNet worsened on both metrics after adding news vectors. PatchTST achieved a lower MSE but a higher MAE. With stride = 48, which approximates one-day-ahead forecasting, adding news vectors generally degraded performance. SARIMAX, Random Forest, TimesNet, and TimeXer showed increases in both MSE and MAE. TimeMixer and PatchTST yielded slightly lower MSE but higher MAE, LSTM achieved less MAE but slightly increased MSE, providing no clear improvement. Overall, the results indicate that the baseline models failed to extract useful signal from the 50-dimensional tfidf news vectors for electricity price forecasting. Under both their default configurations and the stride = 48 setting that simulates one day ahead prediction, adding news vectors offered no benefit or degraded performance.

Without news vectors, increasing the stride from 1 to 48 modestly improves SARIMAX, but degrades Random Forest, LSTM and all state-of-the-art time series models (TimeMixer, TimesNet, TimeXer, PatchTST). With news vectors, the pattern persists: when stride increased, all of the baseline models were worsened. Across settings, baseline errors remain high. The best configuration is LSTM without news vectors (MSE 184355.164, MAE 53.977).

News vectors do not reliably improve baseline models for electricity price forecasting. The state-of-the-art models (TimeMixer, TimesNet, TimeXer, PatchTST) did not surpass LSTM, although each outperformed SARIMAX. Increasing the sliding-window stride reduced their accuracy. Overall, these baselines lack effective mechanisms to integrate 50-dimensional TF-IDF news features for the electricity price forecasting task. These findings expose limitations in the models' ability to incorporate vectorized news in time series forecasting.

Table 3: LLM performance by prompt settings. **Bold** indicates the best result

| Prompt | ChatGPT-4o | | Gemini 1.5 pro | | Meta-Llama-3-8B Instruct | | Mistral-7B-v0.1 | | Qwen2.5-7B-Instruct | |
|---|---|---|---|---|---|---|---|---|---|---|
| | MSE | MAE | MSE | MAE | MSE | MAE | MSE | MAE | MSE | MAE |
| Zero-shot | 105918.758 | 39.213 | 139364.926 | 42.292 | 46100.211 | 51.776 | 46543.277 | 59.821 | 46235.918 | 57.132 |
| Zero-shot CoT | 192582.227 | 58.669 | 146736.599 | 39.999 | 167849.031 | 60.376 | 167394.750 | 72.418 | 167477.656 | 65.005 |
| Few-shot | 111762.174 | 42.431 | 80579.432 | 37.128 | 177259.672 | 59.235 | 176364.484 | 71.259 | 176797.469 | 62.805 |
| Few-shot CoT | 71594.657 | 46.351 | **68214.586** | **35.664** | 98060.195 | 56.024 | 98582.469 | 72.494 | 97638.117 | 56.741 |
| Ablation | 110560.108 | 40.433 | 122651.778 | 38.676 | 122093.227 | 55.707 | 122380.578 | 68.104 | 121982.695 | 62.090 |

**Closed-source and open-source LLMs.** LLMs use one day of history to predict the next day, equivalent to stride 48. Similar to baseline models, news text does not consistently improve forecasts. Despite varied prompt engineering, LLMs are not always better than the ablation that omits news from the prompt. From the Table 3, ChatGPT-4o barely benefit from prompt engineering. Adding few-shot examples or chain-of-thought usually worsens performance. Zero-shot exceeds its ablation, but few-shot and CoT raise both MSE and MAE. Few-shot CoT yields the lowest MSE among engineered prompts, yet it does not improve the overall outcome as it has the second-highest MAE score (46.351). The largest error occurs in zero-shot CoT (MSE = 192582.227; MAE = 58.669).

Open-source LLMs show the same pattern. Meta-Llama-3-8B-Instruct, Mistral-7B-v0.1 and Qwen2.5-7B-Instruct only beat their ablations under zero-shot, while Zero-shot-CoT, few-shot, and few-shot-CoT produce higher MAE; Consistent with ChatGPT-4o, the few-shot CoT setting yields lower MSE compared with ablation but still exhibits high MAE. For the three open-source LLMs, zero-shot is the only prompt that beats their ablations on both MSE and MAE. Zero-shot also gives the lowest MSE and MAE within each model's settings. The other three prompts usually worsen MAE. Few-shot CoT can reduce MSE below the ablation, but it remains far higher than zero-shot. when comparing the open-source llms' performance with each other, Meta-Llama-3-8B Instruct gets the lowest MSE and MAE score in Zero-shot setting. The three LLMs show no meaningful differences in MSE. By MAE, Meta-Llama-3-8B-Instruct is generally lowest across all prompt settings, followed by Qwen2.5-7B-Instruct, then Mistral-7B-v0.1.

Gemini 1.5 Pro behaves differently from all other LLMs. Zero-shot underperforms the ablation. Adding few-shot with chain-of-thought improves both metrics and surpasses the ablation. This setting delivers the best result across models and prompts (MSE = 68,214.586; MAE = 35.664). The MSE nearly halves from zero-shot to few-shot CoT, while the MAE gain is modest, dropping from 42.292 to 35.664. However, relative to the ablation (MAE = 38.676), the MAE improvement remains modest. These results suggest that Gemini 1.5 Pro integrates textual information more effectively for time series forecasting than the other models, yet its capability remains limited. The observed gains are modest and fall short of a robust, general improvement. (The summary of all models' results: appendix I)

**LLM hallucination.** Several hallucination patterns appear in the predicted price series. From Table 4, ChatGPT-4o frequently echoes recent history rather than generating novel forecasts. Prompts without chain of thought exhibit echoing failures: 41.84% in zero shot setting, 21.69% in few shot setting, and 23.27% in the ablation setting. After

Table 4: The percentage of closed-source LLM outputs that exhibit hallucination. **Bold** indicates the highest percentage

| Model | Prompt | Echo Failure % | trivial transformation % | Degenerate copy % | Format Violation % |
|-------|--------|----------------|--------------------------|-------------------|---------------------|
| ChatGPT-4o | Zero shot | **41.84** | 0.55 | 24.42 | 0.05 |
| | Zero shot CoT | 9.15 | 0.08 | 23.03 | 0.47 |
| | Few shot | 21.69 | **10.90** | 3.04 | 0.22 |
| | Few shot CoT | 1.53 | 0.00 | 4.63 | 1.34 |
| | Ablation | 23.27 | 0.38 | **39.43** | 0.00 |
| Gemini 1.5 pro | Zero-shot | 3.04 | 1.29 | 16.70 | 0.00 |
| | Zero-shot CoT | 18.78 | 1.34 | 8.95 | **8.90** |
| | Few shot | 1.15 | 0.58 | 6.79 | 0.00 |
| | Few shot CoT | 0.49 | 0.03 | 4.38 | 2.88 |
| | Ablation | 2.05 | 4.19 | 4.03 | 0.00 |

applying chain of thought, echo failures decreased markedly. The overall rate of trivial transformations was very low, except in the few-shot setting at 10.90%. Degenerate outputs were highest in the ablation, zero shot, and zero shot CoT settings. With few shot prompts, the rate dropped to 3.04%, and adding CoT did not change it much (4.63%). Format violations for ChatGPT-4o were very low across all prompt settings. The ablation had zero violations, while the other four settings showed slightly higher rates.

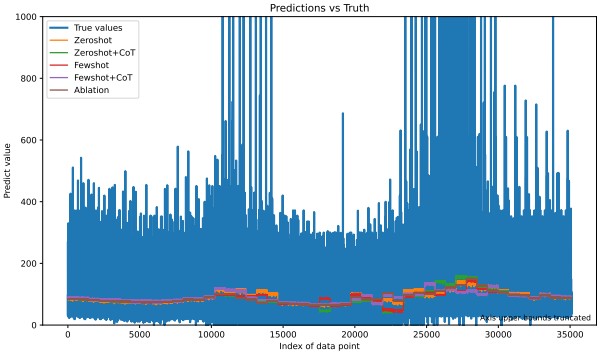

Figure 3: Meta-Llama-3-8B-Instruct predictions vs true values

Figure 4: Mistral-7B-v0.1 predictions vs true values

For Gemini 1.5 Pro, echo failures are generally lower than ChatGPT-4o. The highest rate is 18.78% in zero-shot CoT. CoT raises echo failure in zero shot from 3.04% to 18.78%. In few shot, CoT reduces it from 1.15% to 0.49%. Trivial transformations are also lower than ChatGPT-4o. The highest rate is 4.19% in ablation, and it declines as few shot and CoT are added. Degenerate copies: ablation is lowest. Zero shot is highest at 16.70%. Few shot and CoT reduce it, but none beat ablation. Format violations are zero when CoT is absent. Adding CoT increases error rate. Open source LLM predictions show patterns that differ from those of closed-source models.

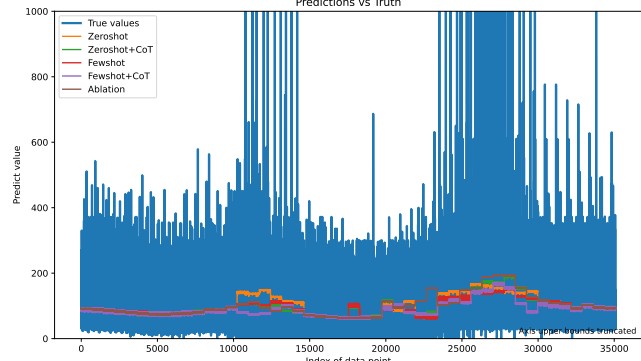

Figure 5: Qwen2.5-7B-Instruct predictions vs true values

The identified hallucination issues for closed-source LLMs are not clearly evident in open-source LLMs' outputs. Instead, their predicted sequences show different patterns (Figure 3, Figure 4 and Figure 5). The y-axis upper bound is truncated to increase the visibility of the LLMs' prediction curves. Open-source LLMs' predicted sequence typically initializes at one of several repeated values and then changes by fixed small steps, suggesting weak conditioning on the context and a bias toward prior numeric templates. Even when the true electricity price series is

highly volatile, open-source LLM forecasts oscillate within a narrow band and do not match the empirical dynamics. Differences across models mainly lie in the amplitude of this band. None of them captures the real volatility pattern, even with varied prompt designs and with vectorized news added.

## 5  CONCLUSION & DISCUSSION

We introduce NSW-EPNEWS, the first benchmark that fuses half-hourly spot prices with curated news summaries, enabling head-to-head evaluation of classical forecasters and state-of-the-art LLMs on a genuinely multimodal electricity-price task. Extensive tests across four prompt regimes reveal that today's leading LLMs and timeseries models lack of abilities in combining news texts in time-series forcasting and the closed-source LLMs remain susceptible to systemic hallucinations—echoing inputs, offset shifts, degenerate repeats, and format failures. These findings expose a substantial gap between present LLMs' capabilities and the reliability required for high-stakes energy forecasting, and they position NSW-EPNEWS as a principled test-bed for future work on prompt design, retrieval grounding, and hallucination mitigation in time-series applications.

**Limitations.**    NSW–EPNews, while providing the first news-augmented benchmark for electricity-price forecasting, is constrained in four respects: (i) all news summaries are produced by GPT-4o, so the impact of alternative LLM summarisers on data fidelity and downstream accuracy remains unknown; (ii) the prompt design space is restricted to four variants (zero/few-shot with or without chain-of-thought), leaving more advanced schemes—such as retrieval-augmented or self-consistency prompts—unexplored; (iii) our hallucination analysis for closed-source LLMs tracks only four surface-level failure modes (echoing, offset shifts, degenerate copies and format violations.), omitting deeper semantic or causal errors; and (iv) empirical evaluation covers five proprietary LLMs (two closed-source LLMs and three open-source LLMs), and fine-tuning alone may be insufficient for open-source LLMs, leaving the behaviour of more closed and open-source LLMs and domain-specialised models to future work.

**Future work.**    Future enhancements will focus on: (i) refining prompt engineering by exploring more sophisticated structures, dynamic guidance, and adaptive prompting strategies; (ii) assessing the suitability of domain-specific lightweight models and AI agent for electricity-price forecasting; (iii) expanding data coverage by incorporating policy bulletins, social-media sentiment, and other heterogeneous textual signals; and (iv) enhancing multimodal fusion—e.g., retrieval-augmented generation (RAG) and graph neural networks to model event relationships—to further boost predictive accuracy; (v) Aim to rigorously tackle hallucination in time series forecasting by further fine-tuning open-source LLMs and analyzing failure modes in greater depth.

**Key Findings.**    Although LLMs excel at natural-language understanding, our experiments show they remain brittle forecasters. Long, multimodal prompts frequently trigger hallucinations—echoed prices, constant offsets, or repeated values—and, under chain-of-thought prompting, some models even announce an "ARIMA step," betraying reliance on canned heuristics rather than genuine reasoning. Injecting news summaries and refined prompts mitigates these pathologies only marginally: LLMs can yield higher MAE and MSE than both the baselines and an ablation that uses only historical prices, and vectorized news typically does not improve their performance. Taken together, the results underline two risks: existing baselines including statistical, deep learning and time series models1 still struggle with extreme volatility, and current LLMs—despite sophisticated prompt engineering—cannot yet deliver reliable, news-aware electricity-price forecasts in high-stakes settings.

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

## A   ETHICS STATEMENT

We comply with the ICLR Code of Ethics and Code of Conduct. This work uses only publicly available, licensed datasets and involves no human subjects or sensitive attributes.

## B   LLM USAGE

We used large language models (LLMs) as experimental subjects and writting assistant, including:

- Summarizing news into structured inputs for experiments;
- Performance testing by doing electricity price predictions under specified prompts and fine-tuning processes (for open-source LLMs).
- Grammar and spelling fixes.

## C   PROMPT SETTINGS

**Zero-shot.**   The zero-shot style prompt includes key contextual information such as one day's worth of historical electricity price data (48 half-hourly points), the summarized news published on that day and the basic time features include the sampling frequency, start and end date of the history data and prediction. The expected output format of the LLM was also explicitly specified in the prompt to ensure consistency and ease of evaluation. This prompt format does not include any examples or explicit reasoning instructions, relying instead on the LLM's generalization ability to produce accurate forecasts for next day's electricity price based solely on the given context.

**Few-shot.**   Building on the zero-shot style prompt, two example question–answer pairs were added to create a few-shot prompt format. This approach is intended to guide the LLM toward generating more consistent and structured responses, while also reducing the likelihood of hallucinations or false responses by providing clear demonstrations of the expected input–output relationship.

**Zero-shot + CoT.**   This approach still follows the zero-shot style but integrated with chain of thought prompting. Chain-of-thought prompting is one of the most commonly used techniques for guiding LLMs toward better reasoning and prediction performance. By providing step-by-step instructions on how to reason about future electricity prices—such as considering historical trends, interpreting relevant news, and accounting for contextual factors—this prompt format encourages the LLM to produce more logical and transparent predictions.

**Few-shot + CoT.**   Both few-shot and chain-of-thought prompting were integrated into the base prompt. Specifically, two illustrative examples containing sample reasoning steps were added to demonstrate how to analyze historical price trends, interpret news content, and arrive at a forecast. This combined approach aims to maximize the reliability of the LLM's output by reducing hallucinations and enhancing forecasting accuracy.

**Ablation Study.**   Before applying the four prompt engineering techniques, we first constructed a zero-shot style prompt that excludes any news data. This prompt allows the LLMs to perform forecasting based solely on the price history, without any additional prompt engineering strategies. This ablation study is designed to highlight the impact of incorporating news and on the forecasting performance of LLMs.

## D   HALLUCINATION AND ERROR EXAMPLES

The hallucination and error outputed by LLMs in the experiment are defined by us as below:

- **Echoing Failure**   Although our prompts explicitly instruct the model to generate forecasts based on an analysis of historical electricity prices and contextual information, the model occasionally echoes segments of the historical data verbatim and presents them as predictions. This behaviour indicates that it is not truly reasoning about future prices, instead, the responses constitute spurious outputs. For each iteration, if ten or more of the LLM's predicted prices are exactly the same as the historical price data, the iteration is considered to have triggered an echoing failure.

Prompt-target Pairs: Zero Shot

[{
  "prompt": "You are an expert in electricity price forecasting. The historical electricity price data is: 26.5, 25.16, 26.07, 22.89, 21.27, 21.25, 20.31, 19.63, 20.01, 24.58, 23.98, 24.75, 26.7, 29.64, 33.09, 35.35, 36.48, 36.7, 42.85, 41.62, 43.34, 49.94, 54.03, 58.12, 54.54, 49.4, 57.33, 55.48, 65.51, 51.47, 76.94, 50.1, 64.88, 95.34, 54.05, 46.33, 42.22, 40.55, 45.75, 44.23, 46.69, 43.23, 38.96, 34.24, 33.71, 34.79, 36.09, 35.98,\nThe start date of historical data was on 06/01/2015 00:00:00. The end date of historical data was on 06/01/2015 23:30:00. The data frequency is 30 minutes per point.\n\nThe news can influence the time series forecasting: 1. The summary of the news \"Recapping prior peaks in NEM-wide demand\" posted at 06-01-2015 03:53:00 PM is: The article discusses …(more texts)\n\nBased on the historical electricity price data and news texts (if given), please predict the electricity price in the next day with a frequency of 30 minutes from 07/01/2015 00:00:00 to 07/01/2015 23:30:00.\n  *Output format**: p1,p2,p3,…,p{expected_count} (no spaces, quotes, newlines, markdown formatting, code block markers or other text).",
  "target": "76.67,56.475,56.475,56.475,53.56,53.085,54.95,54.95,54.95,56.475,57.955,67.8,53.405,35.88,28.17,36.06,35.97,45.06,36.545,35.97,23.975,36.965,55.175,41.675,56.475,59.99,37.26,58.0,56.475,36.06,49.205,35.97,35.97,45.875,61.39,65.195,57.755,71.435,77.805,68.355,66.685,60.37,56.075,64.95,67.595,66.25,68.18,67.585"
  },
  …..(more records)
]

Figure 6: Zero-shot

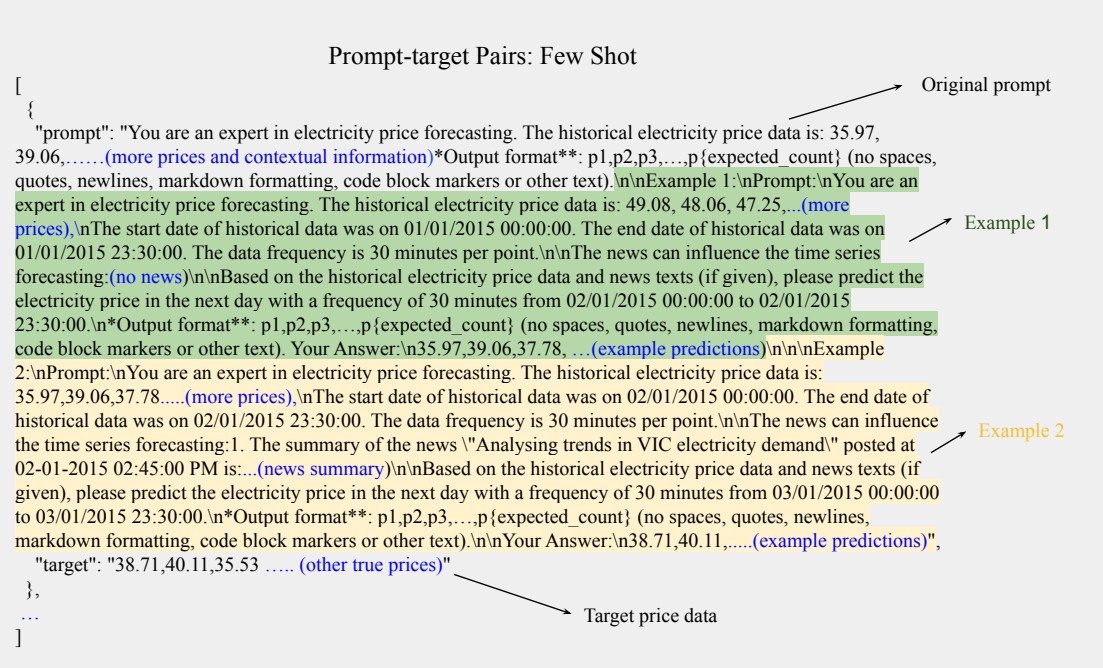

Prompt-target Pairs: Few Shot

Original prompt

[
  {
  "prompt": "You are an expert in electricity price forecasting. The historical electricity price data is: 35.97, 39.06,……(more prices and contextual information)*Output format**: p1,p2,p3,…,p{expected_count} (no spaces, quotes, newlines, markdown formatting, code block markers or other text).\n\nExample 1:\nPrompt:\nYou are an expert in electricity price forecasting. The historical electricity price data is: 49.08, 48.06, 47.25,...(more prices),\nThe start date of historical data was on 01/01/2015 00:00:00. The end date of historical data was on 01/01/2015 23:30:00. The data frequency is 30 minutes per point.\n\nThe news can influence the time series forecasting:(no news)\n\nBased on the historical electricity price data and news texts (if given), please predict the electricity price in the next day with a frequency of 30 minutes from 02/01/2015 00:00:00 to 02/01/2015 23:30:00.\n*Output format**: p1,p2,p3,…,p{expected_count} (no spaces, quotes, newlines, markdown formatting, code block markers or other text). Your Answer:\n35.97,39.06,37.78, …(example predictions)\n\n\nExample 2:\nPrompt:\nYou are an expert in electricity price forecasting. The historical electricity price data is: 35.97,39.06,37.78.....(more prices),\nThe start date of historical data was on 02/01/2015 00:00:00. The end date of historical data was on 02/01/2015 23:30:00. The data frequency is 30 minutes per point.\n\nThe news can influence the time series forecasting:1. The summary of the news \"Analysing trends in VIC electricity demand\" posted at 02-01-2015 02:45:00 PM is:...(news summary)\n\nBased on the historical electricity price data and news texts (if given), please predict the electricity price in the next day with a frequency of 30 minutes from 03/01/2015 00:00:00 to 03/01/2015 23:30:00.\n*Output format**: p1,p2,p3,…,p{expected_count} (no spaces, quotes, newlines, markdown formatting, code block markers or other text).\n\nYour Answer:\n38.71,40.11,.....(example predictions)",
  "target": "38.71,40.11,35.53 ….. (other true prices)"
  },
  …
]

Example 1

Example 2

Target price data

Figure 7: Few-shot

**Prompt-target Pairs: Zero Shot with Chain of Thought**

[
  {
    "prompt": "You are an expert in electricity price forecasting. The historical electricity price data is: 35.97, 39.06, 37.78, 32.41, 32.04, 28.87, 34.22 …(more prices)\nThe start date of historical data was on 02/01/2015 00:00:00. The end date of historical data was on 02/01/2015 23:30:00. The data frequency is 30 minutes per point.\n\nThe news can influence the time series forecasting: 1. The summary of the news \"Analysing trends in VIC electricity demand\" posted at 02-01-2015 02:45:00 PM is: The article analyzes electricity demand…(more texts)\nBased on the historical electricity price data and news texts (if given), please predict the electricity price in the next day with a frequency of 30 minutes from 03/01/2015 00:00:00 to 03/01/2015 23:30:00.\n \nPlease follow these instructions step-by-step:\n\n1. First, perform a detailed analysis of the historical price trend and discuss explicitly:\n  - Patterns observed (increase, decrease, periodicity, spikes, etc.).\n  - Potential impact of the weekday, holidays, or provided news events on future prices.\n  - Your reasoning process on how the above information informs your prediction.\n\n2. After completing your analysis, predict exactly {expected_count} electricity prices.\n\nFinally, output your result in the following exact format—a Python-style list containing two elements, (NO spaces, quotes, newlines, markdown formatting, code block markers or other text):\n\n[\n  \"Your detailed reasoning and analysis goes here.\",\n  \"p1,p2,p3,...,p{expected_count}\"\n]\n",   ← Chain of Thought
    "target": "38.71,40.11,35.53,32.67,31.71,28.69,27.96 …(other true prices)"
  },
  …
]

Figure 8: Zero-shot + CoT

**Prompt-target Pairs: Few Shot with Chain of Thought**

[
  {
    "prompt": "You are an expert in electricity price forecasting. The historical electricity price data is: 35.97, 39.06, 37.78, 32.41, 32.04, 28.87, 34.22 …(more prices and contextual information)\nBased on the historical electricity price data and news texts (if given), please predict the electricity price in the next day with a frequency of 30 minutes from 03/01/2015 00:00:00 to 03/01/2015 23:30:00.\n \nPlease follow these instructions step-by-step:\n\n1. …(CoT instructions)\n\nExample 1:\nPrompt:\nYou are an expert in electricity price forecasting. The historical electricity price data is: 49.08, 48.06, 47.25,…(more prices and contextual information)\n\nBased on the historical electricity price data and news texts (if given), please predict the electricity price in the next day with a frequency of 30 minutes from 02/01/2015 00:00:00 to 02/01/2015 23:30:00.\nPlease follow these instructions step-by-step:\n1….(CoT instructions)\n\nAnswer:\n[\"The historical series first declines then stabilizes with extreme spikes; January 1 …(more reasoning texts), 40.00,39.20, …(more predictions)]\n\nExample 2:\nPrompt:\nYou are an expert in electricity price forecasting. The historical electricity price data is: 35.97,39.06,37.78,…(more prices and contextual information)\n\nBased on the historical electricity price data and news texts (if given), please predict the electricity price in the next day with a frequency of 30 minutes from 03/01/2015 00:00:00 to 03/01/2015 23:30:00.\nPlease follow these instructions step-by-step:\n1….(CoT instructions)\n\nAnswer:\n[\"The series stays mostly in the 30–55 range, with consecutive extreme spikes…(more reasoning texts), \"42.00,41.50,…(more predictions)]
    "target": "38.71,40.11,35.53,32.67,31.71,28.69,27.96 …(other true prices)"
  },
  …]

Figure 9: Few-shot + CoT

- **Trivial Transformation**   Similarly, even when the model avoids repeating the input verbatim,, it sometimes produces forecasts by merely applying a uniform offset to the historical prices. It reveals superficial pattern matching rather than genuine reasoning. If at least 20 of the LLM's predicted values can be obtained by simply adding or subtracting a constant positive number from the historical data, that generation is identified as a trivial transformation case.

- **Degenerate Copying**   In other instances, the model produces the same value repeatedly sometimes for the entire forecast horizon. Although the official electricity-price records occasionally contain stretches of unchanged prices, such flat periods are relatively uncommon. It remains important to record these cases in which the model repeats a single value across its forecasts. We identify a degenerate copying case when the LLM's predicted values are repeating one number for over five times.

- **Format Violation**   LLMs are instructed to output their predicted electricity prices and sometimes along with reasoning steps in specific formats predefined by us. However, in some cases, the models fail to follow the required format, resulting in parsing errors when trying to extract their predicted prices. We record those failures for each LLM when processing prompts. This metric reflects the consistency and reliability of the model's output format, which is a crucial factor in real-world applications where structured post-processing is necessary. If the LLM's answer violates the output format, there will be a parsing error.

### Echoing failure

Prompt: "You are an expert in electricity price forecasting. The historical electricity price data is: 57.98, 57.98, 54.95, 48.91, 41.52, 49.78, 52.22, 46.205, 51.06, 42.26, 36.06, 35.88, 26.485, 4.82, 13.445, 40.395, 0.0, 36.06, 36.06, 44.915, 39.58, 37.2, 54.95, 54.95, 48.575, 28.235, 24.07, 37.125, 30.685, 36.305, 36.06, 31.135, 35.02, 34.745, 36.06, 53.47, 54.95, 51.545, 55.81, 62.74, 56.645, 61.645, 59.23, 56.07, 53.645, 48.915, 49.275, 54.95, …(remaining prompt texts)"

LLM's prediction is simply echoing history data

LLM answer: 57.98, 57.98, 54.95, 48.91, 41.52, 49.78, 52.22, 46.205, 51.06, 42.26, 36.06, 35.88, 26.485, 4.82, 13.445, 40.395, 0.0, 36.06, 36.06, 44.915, 39.58, 37.2, 54.95, 54.95, 48.575, 28.235, 24.07, 37.125, 30.685, 36.305, 36.06, 31.135, 35.02, 34.745, 36.06, 53.47, 54.95, 51.545, 55.81, 62.74, 56.645, 61.645, 59.23, 56.07, 53.645, 48.915, 49.275, 54.95

Figure 10: Echoing Failure

## E   Experiment Settings

### E.1   Baseline models

For baselines, we used SARIMAX (statistical), Random Forest (machine learning), LSTM (deep learning), and state-of-the-art time-series models TimeMixer, TimesNet, TimeXer, and PatchTST. We evaluated each model on two datasets: (1) electricity prices only and (2) electricity prices plus 50-dimensional TF-IDF news vectors. We tested two settings, stride = 1 and stride = 48, for both datasets. Performance was compared using MSE and MAE.

- **LSTM.:** LSTM. We trained a sequence-to-one or sequence-to-multi LSTM regressor on half-hourly price data with optional exogenous features. Data were split by time into 70% train, 10% validation, and 20% test. We standardized all features with a StandardScaler fit on the training slice only, then applied the transform to validation and test. Supervised samples used a lookback window L=48 and forecasting horizon H. Training,

## Trival Transform

Prompt: You are an expert in electricity price forecasting. The historical electricity price data is: 57.98, 57.98, 54.95, 48.91, 41.52, 49.78, 52.22, 46.205, 51.06, 42.26, 36.06, 35.88, 26.485, 4.82, 13.445, 40.395, 0.0, 36.06, 36.06, 44.915, 39.58, 37.2, 54.95, 54.95, 48.575, 28.235, 24.07, 37.125, 30.685, 36.305, 36.06, 31.135, 35.02, 34.745, 36.06, 53.47, 54.95, 51.545, 55.81, 62.74, 56.645, 61.645, 59.23, 56.07, 53.645, 48.915, 49.275, 54.95, …(remaining prompt texts)"          LLM's answer is simply transforming historical data by 1.0

LLM answer: 58.98, 58.98, 55.95, 49.91, 42.52, 50.78, 53.22, 47.205, 52.06, 43.26, 37.06, 36.88, 27.485, 5.82, 14.445, 41.395, 1.0, 37.06, 37.06, 45.915, 40.58, 38.2, 55.95, 55.95, 49.575, 29.235, 25.07, 38.125, 31.685, 37.305, 37.06, 32.135, 36.02, 35.745, 37.06, 54.47, 55.95, 52.545, 56.81, 63.74, 57.645, 62.645, 60.23, 57.07, 54.645, 49.915, 50.275, 55.95

Figure 11: trivial transformation

## Degenerate Copy

Prompt: "You are an expert in electricity price forecasting. The historical electricity price data is: 57.98, 57.98, 54.95, 48.91, 41.52, 49.78, 52.22, 46.205, 51.06, 42.26, 36.06, 35.88, 26.485, 4.82, 13.445, 40.395, 0.0, 36.06, 36.06, 44.915, 39.58, 37.2, 54.95, 54.95, 48.575, 28.235, 24.07, 37.125, 30.685, 36.305, 36.06, 31.135, 35.02, 34.745, 36.06, 53.47, 54.95, 51.545, 55.81, 62.74, 56.645, 61.645, 59.23, 56.07, 53.645, 48.915, 49.275, 54.95, …(remaining prompt texts)"    LLM's answer is simply repeating one number many times

LLM answer: 57.98, 57.98, 57.98, 57.98, 57.98, 57.98, 57.98, 57.98, 57.98, 57.98, 57.98, 57.98, 57.98, 57.98, 57.98, 57.98, 57.98, 57.98, 57.98, 57.98, 57.98, 57.98, 57.98, 57.98, 57.98, 57.98, 57.98, 57.98, 57.98, 57.98, 57.98, 57.98, 57.98, 57.98, 57.98, 57.98, 57.98, 57.98, 57.98, 57.98, 57.98, 57.98, 57.98, 57.98, 57.98, 57.98, 57.98

Figure 12: Degenerate Copy

## Format Violation

Prompt: "You are an expert in electricity price forecasting. The historical electricity price data is: 57.98, 57.98, 54.95, 48.91, 41.52, 49.78, 52.22, 46.205, 51.06, 42.26, 36.06, 35.88, 26.485, 4.82, 13.445, 40.395, 0.0, 36.06, 36.06, 44.915, 39.58, 37.2, 54.95, 54.95, 48.575, 28.235, 24.07, 37.125, 30.685, 36.305, 36.06, 31.135, 35.02, 34.745, 36.06, 53.47, 54.95, 51.545, 55.81, 62.74, 56.645, 61.645, 59.23, 56.07, 53.645, 48.915, 49.275, 54.95, …(background information), *Output format**: p1,p2,p3,…,p{expected_count} (no spaces, quotes, newlines, markdown formatting, code block markers or other text).",

Outputted texts, which violates the format requirement

LLM answer: "Based on the provided historical load data, I will use … (more text)"

Figure 13: Format Violation

## Format Violation (For CoT response)

Prompt: "You are an expert in electricity price forecasting. The historical electricity price data is: 57.98, 57.98, 54.95, 48.91, 41.52, 49.78, 52.22, 46.205, 51.06, 42.26, 36.06, 35.88, 26.485, 4.82, 13.445, 40.395, 0.0, 36.06, 36.06, 44.915, 39.58, 37.2, 54.95, 54.95, 48.575, 28.235, 24.07, 37.125, 30.685, 36.305, 36.06, 31.135, 35.02, 34.745, 36.06, 53.47, 54.95, 51.545, 55.81, 62.74, 56.645, 61.645, 59.23, 56.07, 53.645, 48.915, 49.275, 54.95, …(background information and CoT instructions), output your result in the following exact format—a Python-style list containing two elements:\n\n[\n    \"Your detailed reasoning and analysis goes here.\",\n    \"p1,p2,p3,...,p{expected_count}\"\n]

The answer is not a Python-style List, which violates the requirement of formatting

LLM answer: ```Based on the provided historical load data, I will use … (more text)```

Figure 14: Format Violation (For CoT answers)

validation and test samples used strde s = 1 or 48. The model had an LSTM backbone with hidden size 256, two layers, dropout 0.2, and a linear head that maps the last hidden state to H outputs for multi-output or one output for single-step. We optimized with Adam at learning rate 1e-3, batch size 256, for up to 40 epochs with early stopping patience 5 and gradient clipping at 1.0. The device was CUDA if available, otherwise CPU.

- **SARIMAX.**: We evaluated SARIMAX with optional exogenous regressors under rolling or expanding windows. The dataset was split by time into 70% train, 10% validation, and 20% test. For each window start t we fit SARIMAX(p,d,q) on y[0:t] or on the last window size points if rolling, then forecast H steps using the aligned exogenous slice. The evaluation target was the last step at t+H-1. We advanced t by the user-set stride s(1 or 48). The window size and horizon was all setted to 48.

- **Random Forest**: Each run uses a sliding window of 48 historical steps to predict the next 48 steps. We test two sampling strides, stride = 1 and stride = 48, yielding four configurations per dataset–stride pair. The Random Forest uses 400 trees (n_estimators = 400), a maximum depth of 20, and n_jobs = -1 to parallelize across CPU cores. Data are split chronologically with 70% for training, 10% for validation, and the remainder for testing.

- **PatchTST.** Two variants were run. Dataset with electricity prices and news vectors used 51 input channels (price + 50 TFIDF) and predicted 48 steps with $L = 48$ and $H = 48$. Encoder layers 1. Decoder layers 1. Heads 2. Factor 3. `enc_in=51`, `dec_in=51`, `c_out=51`. Dataset without news vectors used `enc_in=1`, `dec_in=1`, `c_out=1`. Dataloader accepted a configurable sliding stride.

- **TimeMixer.** Dataset with electricity prices and news vectors used 51 channels with $L = 48$, $H = 48$, and label length 0. Encoder layers 2. $d\_model = 16$. $d\_ff = 32$. Downsampling layers 3. Window 2. Method `avg`. Epochs 10. Patience 10. Learning rate 0.01. Batch size 128. `enc_in=51`, `c_out=51`. Dataset without news vectors used 1 channel with label length 48 and the same core hyperparameters, with `enc_in=1`, `c_out=1`. The dataloader supported a user stride.

- **TimesNet.** Dataset with electricity prices and news vectors used 51 channels with $L = 48$, label length 48, and $H = 48$. Encoder layers 2. Decoder layers 1. Factor 3. $d\_model = 16$. $d\_ff = 32$. `top_k`=5. `enc_in=51`, `dec_in=51`, `c_out=51`. Dataset without news vectors used 1 channel with `enc_in=1`, `dec_in=1`, `c_out=1`. The dataloader accepted a sliding stride.

- **TimeXer.** Dataset with electricity prices and news vectors used 51 price+news channels with $L = 48$, label length 48, and $H = 48$. Encoder layers 1. Factor 3. `d_model = 256`. Batch size 4. `enc_in=51`, `dec_in=51`, `c_out=53`. Dataset without news vectors used 1 channel with `enc_in=1`, `dec_in=1`, `c_out=1`. The only code change was enabling variable sliding stride in the dataloader.

### E.2 LARGE LANGUAGE MODELS

**Closed-source LLMs.** We selected two state-of-the-art closed-source LLMs, ChatGPT-4o(OpenAI, 2024) and Gemini 1.5 Pro(Reid et al., 2024). They will be evaluated using the constructed prompt–target pairs. Each LLM was tested with zero-shot, few-shot, zero-shot+CoT, few-shot+CoT prompt-engineering strategies. During evaluation, prompts were fed to the models via iterative API calls. The generated responses will be compared with the target values to calculate error scores. In addition, the generated responses will also be compared against the provided historical price data to identify instances of hallucinations and erroneous outputs. This process includes checking whether the generated responses follow the required output format. If the format is correct, we further examine whether the responses simply echo the historical prices, apply a fixed[offset to them, or consist of a repeated single value throughout the forecast sequence. Then LLMs' predicted price sequence will be extracted and scored against the true prices using MAE and MSE.

**Open-source LLMs.** We fine-tuned open-source LLMs with a lightweight regression head using LoRA. The chosen base models were Meta-Llama-3-8B-Instruct, Mistral-7B-v0.1 and Qwen-2.5-7B-Instruct, loaded in 4-bit. We trained for 5 epochs with batch size 1 and learning rate 1e-4. The forecasting horizon was 48 and the sequence max length was 3000. Data were JSON/JSONL with fields prompt and target; targets were parsed as 48-dimensional float vectors. A special token "<PRED>" marked the prediction location; The loss was mean squared error. Optimization used AdamW with a cosine schedule and a warmup ratio of 0.1; weight decay was 0.0. Training proceeded by epochs rather than max steps; LoRA used r=8, alpha=16, dropout=0.05, and targeted q_proj and v_proj. Inference batched prompts and reused the "<PRED>" mechanism. Evaluation reported MSE and MAE. Computation ran on CUDA with bfloat16 when available, otherwise float32.

## F NEWS ARTICLE EXAMPLE FROM WATTCLARITY

- On Wednesday 27th November 2024 2:48 PM, a news titled with "NSW Prices spike early ahead of forecasts on afternoon of 27th November" posted: "A quick note to record that Energy price in NSW has already spiked to up near the market price cap of $17500 as early as 14:30 NEM time, seen here in Forecast Convergence grid view. Pricing outcomes on high demand days like this are clearly driven by a number of complex factors, but increasingly we are finding value in the overall story being told by our Congestion Map prototype."(Kent, 2024)

- On Friday 8th November 2024 7:03 PM, a news titled with "A much shorter run of evening volatility in QLD and NSW on Friday 8th November 2024" posted " It was more humid on the walk home from the Brisbane office late this afternoon ... but at least there were not as many SMS pings as there were for price alerts on Thursday 7th November 2024, with the following being the capture of prices in any region above $1,000/MWh...."(McArdle, 2024)

## G EVALUATION METRICS

For both baseline models and LLMs, we compare their predicted prices with thte true prices one by one and calcualte the standard metrics. Note that LLMs may occasionally produce slightly more or fewer predicted price values than expected. In such cases, any extra predictions are discarded. Missing values are initially filled with null placeholders and later excluded, along with their corresponding ground truth values, during the calculation of error metrics. We used four standard evaluation metrics: mean squared error (MSE), mean absolute error (MAE), root mean squared error (RMSE), and mean absolute percentage error (MAPE).

- **Mean Squared Error (MSE)**: MSE is sensitive to large errors and thus emphasizes the impact of extreme values in electricity price forecasting. It is calculated by averaging the squares of the differences between predicted and actual prices.

- **Mean Absolute Error (MAE)**: MAE reflects the average absolute difference between the predicted and actual prices. It provides a straightforward measure of the typical prediction error in Australian dollars and is less sensitive to outliers compared to MSE and RMSE.

The formulas are shown below, where $y_i$ denotes the $i$-th true value, $\hat{y}_i$ denotes the corresponding predicted value, and $N$ is the total number of samples.

$$\text{MAE} = \frac{1}{N}\sum_{i=1}^{N}|\hat{y}_i - y_i| \qquad \text{MSE} = \frac{1}{N}\sum_{i=1}^{N}(\hat{y}_i - y_i)^2$$

## H NEWS CLASSIFYING PROMPT

The prompt that used by ChatGPT-4o to summarize news including following key points.

- **Role assignment.** The prompt begins with *"You are an expert in electricity market analysis"*. This persona priming has two effects: (i) it biases the model toward domain-relevant vocabulary and causal reasoning (e.g. linking turbine failure to reserve scarcity), and (ii) it reduces generic, copy-editing style hallucinations by anchoring generation in an expert voice.

- **Classification criteria.** We supply an explicit three-level taxonomy—LEVEL 1 catastrophic outages or fuel shortages, LEVEL 2 operational or price signals, LEVEL 3 policy or sentiment items. Embedding the taxonomy inside the prompt forces the model to issue a categorical relevance label, enabling us to stratify the benchmark and later test whether forecasters place higher weight on LEVEL 1 events than on routine policy news.

- **Key attributes.** The prompt enumerates ten named fields (*timeframe of impact*, *root cause*, *affected region*, etc.). These slots compel GPT-4o to transform unstructured prose into a fixed schema that is readily ingested by downstream models and analytic scripts. When an attribute is absent in the article, the model must output `Unknown`, making missingness explicit and avoiding silent information leakage.

- **Summary rules.** Finally, a rigid output template (length $\leq$ 30,000 characters, keyword–value pairs, date self-verification) standardises length and formatting. This constraint not only simplifies post-processing but also limits free-form text that could introduce syntactic noise or hallucinated details.

# I  EXPERIMENT RESULTS

Table 5: All models' performance summary

| Model | Settings | MSE | MAE |
|---|---|---|---|
| SARIMAX | Electricity prices (stride = 1) | 522449.897 | 87.532 |
| | Electricity prices with (stride = 48) | 477957.310 | 84.236 |
| | With 50-TF-IDF news vectors (stride = 1) | 516962.132 | 87.367 |
| | With 50-TF-IDF news vectors (stride = 48) | 780992.609 | 108.031 |
| Random Forest | Electricity prices (stride = 1) | 207472.724 | 65.759 |
| | Electricity prices with (stride = 48) | 232346.160 | 74.378 |
| | With 50-TF-IDF news vectors (stride = 1) | 220005.819 | 70.073 |
| | With 50-TF-IDF news vectors (stride = 48) | 223198.027 | 74.338 |
| LSTM | Electricity prices (stride = 1) | 184355.164 | 53.977 |
| | Electricity prices with (stride = 48) | 186491.115 | 72.272 |
| | With 50-TF-IDF news vectors (stride = 1) | 187373.364 | 54.863 |
| | With 50-TF-IDF news vectors (stride = 48) | 189332.058 | 64.728 |
| TimeMixer | Electricity prices (stride = 1) | 195509.203 | 64.728 |
| | Electricity prices with (stride = 48) | 218554.609 | 73.969 |
| | With 50-TF-IDF news vectors (stride = 1) | 208713.172 | 61.341 |
| | With 50-TF-IDF news vectors (stride = 48) | 217606.344 | 73.998 |
| TimesNet | Electricity prices (stride = 1) | 208088.125 | 65.715 |
| | Electricity prices with (stride = 48) | 232091.781 | 78.685 |
| | With 50-TF-IDF news vectors (stride = 1) | 217206.641 | 69.994 |
| | With 50-TF-IDF news vectors (stride = 48) | 250649.750 | 84.495 |
| TimeXer | Electricity prices (stride = 1) | 203066.656 | 60.089 |
| | Electricity prices with (stride = 48) | 234809.594 | 79.001 |
| | With 50-TF-IDF news vectors (stride = 1) | 203789.875 | 59.423 |
| | With 50-TF-IDF news vectors (stride = 48) | 236026.859 | 79.311 |
| PatchTST | Electricity prices (stride = 1) | 209318.531 | 59.154 |
| | Electricity prices with (stride = 48) | 235771.938 | 80.740 |
| | With 50-TF-IDF news vectors (stride = 1) | 207297.938 | 65.475 |
| | With 50-TF-IDF news vectors (stride = 48) | 233728.422 | 83.848 |
| ChatGPT-4o | Prompt: Zero-shot | 105918.758 | 39.213 |
| | Prompt: Zero-shot CoT | 192582.227 | 58.669 |
| | Prompt: Few-shot | 111762.174 | 42.431 |
| | Prompt: Few-shot CoT | 71594.657 | 46.351 |
| | Prompt: Ablation | 110560.108 | 40.433 |
| Gemini 1.5 pro | Prompt: Zero-shot | 139364.926 | 42.292 |
| | Prompt: Zero-shot CoT | 146736.599 | 39.999 |
| | Prompt: Few-shot | 80579.432 | 37.128 |
| | Prompt: Few-shot CoT | 68214.586 | 35.664 |
| | Prompt: Ablation | 122651.778 | 38.676 |
| Meta-Llama-3-8B-Instruct | Prompt: Zero-shot | 46201.648 | 48.932 |
| | Prompt: Zero-shot CoT | 168244.281 | 60.376 |
| | Prompt: Few-shot | 177204.344 | 57.644 |
| | Prompt: Few-shot CoT | 97878.875 | 58.127 |
| | Prompt: Ablation | 122352.852 | 54.885 |
| Mistral-7B-v0.1 | Prompt: Zero-shot | 46438.898 | 59.150 |
| | Prompt: Zero-shot CoT | 167689.594 | 71.913 |
| | Prompt: Few-shot | 176334.844 | 70.357 |
| | Prompt: Few-shot CoT | 98673.430 | 73.318 |
| | Prompt: Ablation | 122593.328 | 70.186 |
| Qwen-2.5-7B-Instruct | Prompt: Zero-shot | 46235.918 | 57.132 |
| | Prompt: Zero-shot CoT | 167477.656 | 65.005 |
| | Prompt: Few-shot | 176797.469 | 62.805 |
| | Prompt: Few-shot CoT | 97638.117 | 56.741 |
| | Prompt: Ablation | 121982.695 | 62.090 |