# OpenReview forum: "NSW-EPNEWS: A NEWS-AUGMENTED BENCHMARK FOR ELEC- TRICITY PRICE FORECASTING WITH LLMS"
_ICLR.cc/2026/Conference — Submitted to ICLR 2026_

### Official Review · Reviewer_oc8m · 2025-10-25

**Soundness:** 2
**Presentation:** 2
**Contribution:** 1
**Rating:** 2
**Confidence:** 3

**Summary:**

This paper introduces NSW-EPNews, a benchmark combining electricity prices from New South Wales with news summaries from WattClarity to evaluate how traditional forecasting models and LLMs use textual information. The dataset is used to test models from various LLMs under various prompt templates (zero/few-shot, with/without CoT). Experiments show that adding news features does not consistently improve forecasting and often degrades performance. LLMs exhibit frequent hallucinations.

**Strengths:**

- The benchmark is new to the community.
- The authors conducted rigorous evaluation across statistical, deep learning, and LLM baselines under consistent data splits and metrics.
- Experiment details, including preprocessing (scraping, summarization, down-sampling, and prompt design) are clear.

**Weaknesses:**

- The paper’s central empirical finding—that incorporating news features does not improve and often even degrades electricity price forecasting, is not surprising and inconclusive. This result could stem from various factors including: (a) the actual irrelevance of the selected textual data or (b) weaknesses in how text was represented and integrated. Without stronger evidence or ablation analyses, the conclusion may be overgeneralized.
- Specifically, the text processing relies primarily on TF-IDF vectors and GPT-generated summaries, without sufficient justification or validation of these representations. As a result, it remains unclear whether the observed degradation reflects inherent limitations of textual data.
- The considered dataset, while useful, is geographically limited to a single region (NSW–Australia), which restricts the generalizability of the findings.
- In terms of methodological contribution, the work functions largely as a benchmarking and evaluation study, and thus technical novelty is limited.
- While the dataset offers some value to the community, framing it as a “benchmark for multimodal LLM reasoning” feels overstated given that (a) the scope of the dataset is limited and (b) text modality is not expressive enough.

**Questions:**

Please refer to the weaknesses above.

---

### Official Review · Reviewer_UC83 · 2025-10-31

**Soundness:** 2
**Presentation:** 2
**Contribution:** 2
**Rating:** 2
**Confidence:** 4

**Summary:**

The paper presents a dataset for electricity price forecasting using multimodal data, namely, text and time series data. The paper provides the initial benchmark results for electricity forecasting using this multimodal data. Initial benchmark results in the paper demonstrate that there is no significant advantage of using text and time series data as compared to traditional models.

**Strengths:**

1. Paper is clear and well presented.
2. Dataset and the results are clearly mentioned and the potential use of this dataset has been described in detail for electricity domain specialists to leverage and develop new algorithms.

**Weaknesses:**

Major:
1. There are richer datasets on multimodal data like finance and wind turbines in the open source domain already. Example:
https://huggingface.co/datasets/Wenyan0110/Multimodal-Dataset-Image_Text_Table_TimeSeries-for-Financial-Time-Series-Forecasting

 It is not clear what are the major advantages of using this dataset beyond testing it for the electricity domain. The paper seems better placed for an electricity domain focused conference.

Minor:
1. Reference error : yielding sizeable accuracy gains via in-context learning (?)
2. I might have misunderstood this point and so some clarification can help. Even if open-sourced, it is not clear how this dataset can be used by the community as the licensing aspects of this dataset can be tricky since the dataset is not generated by the authors or by scraping open-source  data. It is generated by scraping the data from a private source (WattClarity(Global-Roam Pty Ltd, 2025)).
3. Inconsistency in citations. Some places have a space before the citation and some places have no space: prediction quality (Wei et al., 2023a) vs signals(Menéndez Medina and Heredia Álvaro, 2024)

**Questions:**

1. What additional aspects can be leveraged from this dataset to develop new multimodal algorithms as compared to other existing multimodal datasets (like finance etc).
2. Have the news summaries been validated for hallucination and accuracy? How was this done? On a related note, what does “curated” mean in the context of this paper? How was it curated? (I might have missed this in the paper)

**Details Of Ethics Concerns:**

Even if open-sourced, it is not clear how this dataset can be used by the community as the licensing aspects of this dataset can be tricky since the dataset is not generated by the authors or by scraping open-source  data. It is generated by scraping the data from a private source (WattClarity(Global-Roam Pty Ltd, 2025)). I might have missed this aspect in the paper but asked the authors for clarification.

---

### Official Review · Reviewer_BPUT · 2025-11-01

**Soundness:** 3
**Presentation:** 4
**Contribution:** 2
**Rating:** 4
**Confidence:** 4

**Summary:**

The paper introduces a new benchmark dataset "NSW-EPNews" that includes 175K half-hourly spot electricity prices with curated market-news text summaries.This dataset consist of 3.6K multi-modal prompt-output pairs to pave the way for multi-modal evaluation with LLMs. The authors evaluate both time-series only models (SARIMAX, RandomForest, LSTM, PatchTST etc) with and without news augmented features and also LLM based models (ChatGPT-4o, Gemini 1.5 pro etc) on the 48 step ahead forecasting (next day). The main finding of the paper is that the models augmented with curated market news did not provide significant gains and even degraded w.r.t the time-series only models.

**Strengths:**

The main strengths of the paper include,
1. Introduced a novel multi-modal benchmark dataset consisting of text-time series data points for electricity price forecasting.
2. Conducted thorough evaluation of the time-series only models (SARIMAX, RandomForest, LSTM, PatchTST etc) with and without news augmented features.
3. Performed thorough investigation on the LLM capababilities of popular open/closed source LLM models in utilizing the multi-model text-time series inputs to forecast 48 step ahead electricity prices.
4. Conducted hallucination analysis of the LLM models for pre-defined failure modes (echoing, trivial transformations, degenerate copying, format violations) is a useful insight for the community.
5. The paper is well written and easy to follow. The authors provided clear description of the data curation process, model evaluation setup etc.

**Weaknesses:**

The main weakness of the paper include,
1. The paper mentions the central finding that news information doesn't help with improving the forecasts for both time-series only models and LLM-based models but doesn't explain the reasons. They also doesn't attribute if the performance degradation is due to the textual context not being correlated to price changes (or) the models evaluated doesn't have the capability to use the textual context to improve the forecasts.
2. The authors use TF-IDF to create vector representation for textual context but that is an very old method and might not capture the true semantic meaning of the textual context. So, trying recent approaches like fast-text embeddings, BERT-style encoders to extract vectorized features might be important to have a fair conclusion.
3. The baseline models are not designed/modified to consume both time series and textual information. So, it might not be fair to conclude the news information is not helping with the performance of electricity price forecasting. I would like to see any baseline method that supports consuming multi-modal input better than the trivial concatenation approaches mentioned in the paper.

**Questions:**

1. Any reason why the problem is formulated as using only last 48 time step electricity prices to forecast the next 48 time step electricity prices? Why not use more than last 1 day (for eg, last 14 days (or) last 30 days) as electricity prices might have seasonality associated with them which cannot be captured using only the last day prices.
2. How was the quality of GPT-4o summaries validated as part of data curation step? Are the different summarizers compared/validated?
3. Given negative results on the usage of news information for electricity price forecasting, what are the takeaways for the community? Should they avoid news entirely or is better integration possible?
4. Can you try using better mechanisms to encode text inputs other than TF-IDF (like fast-text embeddings, BERT-style encodings etc) and validate the conclusion?
5. Can you obtain the results of baseline multi-modal models that non-trivially consume both text and time series instead of trivial approaches mentioned in the paper?

---

### Official Review · Reviewer_HErU · 2025-11-01

**Soundness:** 2
**Presentation:** 2
**Contribution:** 2
**Rating:** 2
**Confidence:** 3

**Summary:**

This paper introduces a new-augmented benchmark for electricity price forecasting with LLMs. The benchmark data is sourced by scraping web data and preprocessed with GPT-4o, and evaluation is done with various prompting strategies. The experiments show that existing LLMs do not benefit from the news information and often perform worse when exposed to it.

**Strengths:**

- The paper provides detailed information on how the benchmark is constructed and on the evaluation procedure.
- Experiments are done with many baselines methods and various LLMs.
- Demonstrates the surprising finding that LLMs do not benefit from the news information despite their text understanding.

**Weaknesses:**

- It is unclear that the proposed benchmark can measure anything useful. The implicit assumption is that the news information should help capable LLM forecasters that can understand text, yet there the paper provides no evidence that the news indeed provides useful signal that can be leveraged to make better forecasts. In other words, the result that LLMs prediction don't improve with the provided news information may say more about the benchmark than about the LLM forecasters. Indeed prior work [1] has shown that LLM forecasters indeed benefit from text conditioning when that text is informative of the prediction task.
- The restriction to the electricity price data results in an unnecessarily narrow domain, where news information does not play a distinctive role than other domains for time-series forecasting.

[1] Requeima, James, et al. "Llm processes: Numerical predictive distributions conditioned on natural language." Advances in Neural Information Processing Systems 37 (2024)

**Questions:**

- How can we distinguish, in a quantitative way, whether the news text provides actual useful information for forecasting but the LLMs fail make use of it, or that there is simply little useful information in the news for making better forecasts, e.g. due to very low signal-to-noise ratio.
- How do you reconcile your findings with prior work that demonstrates LLM forecasters can benefit from informative text conditioning?

---

### Meta-Review · Area_Chair_jhBX · 2026-01-05

**Summary:**

This paper has been assessed by four knowledgeable reviewers who all voted to reject it (3 straight rejection scores, one marginal reject). They were concerned that the paper’s main claim, that news features do not improve electricity price forecasting, is weak and inconclusive, likely reflecting limitations in text representation and model design choices rather than true irrelevance of news data.
Moreover, narrow dataset, lack of ablation studies, minimal methodological novelty, and appearing overstated benchmark claims, further undermine generalizability and impact of this paper. In summary, it is not ready for inclusion in the program of ICLR.

**Reviewer Concerns:**

The authors have not provided a rebuttal.

**Reviewer Scores:**

I do not believe the scores would change the assessment of the paper, perhaps the reviewer giving it a 4 would reduce their rating.

---

### Decision · Program_Chairs · 2026-01-26

Reject